# Impact of leakage during HFC-125 production on the increase in HCFC-123 and HCFC-124 emissions

Luke M. Western[1,2], Stephen Bourguet[3], Molly Crotwell[4,5], Lei Hu[5], Paul B. Krummel[6], Hélène De Longueville[1], Alistair J. Manning[7], Jens Mühle[8], Dominique Rust[1], Isaac Vimont[5], Martin K. Vollmer[9], Minde An[2], Jgor Arduini[10], Andreas Engel[11], Paul J. Fraser[6], Anita L. Ganesan[12], Christina M. Harth[8], Chris Lunder[13], Michela Maione[10], Stephen A. Montzka[5], David Nance[4,5], Simon O'Doherty[1], Sunyoung Park[14,15], Stefan Reimann[9], Peter K. Salameh[16], Roland Schmidt[8], Kieran M. Stanley[1], Thomas Wagenhäuser[11], Dickon Young[1], Matt Rigby[1], Ronald G. Prinn[2], and Ray F. Weiss[8]

[1]School of Chemistry, University of Bristol, Bristol, United Kingdom
[2]Center for Sustainability Science and Strategy, Massachusetts Institute of Technology, Cambridge, MA, USA
[3]Earth Commons, Georgetown University, Washington, DC, USA
[4]Cooperative Institute for Research in Environmental Sciences, University of Colorado, Boulder, CO, USA
[5]NOAA Global Monitoring Laboratory, Boulder, CO, USA
[6]CSIRO Environment, Aspendale, Victoria, Australia
[7]Hadley Centre, Met Office, Exeter, UK
[8]Scripps Institution of Oceanography, University of California San Diego, La Jolla, CA, USA
[9]Empa – Laboratory for Air Pollution / Environmental Technology, Dübendorf, Switzerland
[10]Department of Pure and Applied Sciences, University of Urbino, Urbino, Italy
[11]Institute for Atmospheric and Environmental Science, Goethe University Frankfurt, Frankfurt am Main, Germany
[12]School of Geographical Sciences, University of Bristol, Bristol, UK
[13]NILU, Kjeller, Norway
[14]Department of Oceanography, Kyungpook National University, Daegu, Republic of Korea
[15]Kyungpook Institute of Oceanography, Kyungpook National University, Daegu, Republic of Korea
[16]GC Soft Inc., Carlsbad, CA, USA

**Correspondence:** Luke M. Western (lwestern@mit.edu)

**Abstract.** Hydrochlorofluorocarbons (HCFCs) are ozone-depleting substances whose production and consumption have been phased out under the Montreal Protocol in non-Article 5 (mainly developed) countries and are currently being phased out in the rest of the world. Here, we focus on two HCFCs, HCFC-123 and HCFC-124, whose emissions are not decreasing globally in line with their phase-out. We present the first measurement-derived estimates of global HCFC-123 emissions (1993-2023) and updated HCFC-124 emissions for 1978-2023. Around 5 Gg yr$^{-1}$ of HCFC-123 and 3 Gg yr$^{-1}$ of HCFC-124 were emitted in 2023. Both HCFC-123 and HCFC-124 are intermediates in the production of HFC-125, a non-ozone-depleting hydrofluorocarbon (HFC) that has replaced ozone-depleting substances in many applications. We show that it is possible that the observed global increase in HCFC-124 emissions could be entirely due to leakage from the production of HFC-125, provided that its leakage rate is around 1% by mass of HFC-125 production. Global emissions of HCFC-123 have not decreased despite its phase-out for production under the Montreal Protocol, and its use in HFC-125 production may be a contributing factor to this. Emissions of HCFC-124 from western Europe, the USA and East Asia have either fallen or not increased since 2015 and together cannot explain the entire increase in the derived global emissions of HCFC-124. These findings add to the

growing evidence that emissions of some ozone-depleting substances are increasing due to leakage and improper destruction during fluorochemical production.

## 1  Introduction

The production of most long-lived ozone-depleting substances (ODSs) has been or is currently being phased out under the Montreal Protocol on Substances that Deplete the Ozone Layer. Levels of most chlorofluorocarbons (CFCs) and hydrochlorofluorocarbons (HCFCs) are falling in the atmosphere (Laube and Tegtmeier, 2023; Western et al., 2024). As a result, the Antarctic Ozone Hole is projected to recover to a state similar to that in the 1980s by the 2060s (Hassler and Young, 2023).

There has been an overall decline in emissions of ODSs, yet there have been several instances when emissions of individual ODSs have increased despite a production phase-out or phase-down. Emissions of CFC-11 ($CCl_3F$) increased for several years due to unreported production and use (Montzka et al., 2021, 2018; Rigby et al., 2019; Park et al., 2021). The emissions of HCFC-141b ($CH_3CCl_2F$), a replacement for CFC-11, increased between 2017 and 2021 (Western et al., 2022). The reason for the increase in emissions is unclear but is likely attributable, at least in part, to emissions during the disposal of appliances at their end of life. Various other CFCs and HCFCs have had persistent or increasing emissions (Adcock et al., 2018; Lickley et al., 2020; Vollmer et al., 2021, 2018; Western et al., 2023). The increase in the emissions of these CFCs and HCFCs has been largely attributed to their involvement in the production of other chemicals, most notably hydrofluorocarbons (HFCs) (Adcock et al., 2018; Kloss et al., 2014; Vollmer et al., 2015, 2018, 2021; Western et al., 2023), which is allowed under the Montreal Protocol. These ODSs are still produced as chemical feedstock (raw) materials, as intermediate chemicals (a necessary chemical step during the production process) and by-products (an often-unwanted result of the production process). Leakage during the production chain is likely the cause of their emissions (Daniel and Reimann, 2023; TEAP, 2024). A change in the emissions of substances involved in the production of HFCs can be indicative of changes in production volumes, relative changes in chemical production pathways, and/or leakage rates during the production of HFCs. Here we show that emissions are not falling in line with the production phase-out for two additional HCFCs, HCFC-123 (2,2-dichloro-1,1,1-trifluoroethane, $C_2HCl_2F_3$) and HCFC-124 (1-chloro-1,2,2,2-tetrafluoroethane, $C_2HClF_4$), which are largely phased out for dispersive usages (i.e., non-fluorochemical production).

HCFCs were used as replacements for CFCs as they have a smaller impact on stratospheric ozone depletion due to their shorter lifetimes in the atmosphere. HCFC-124 has an atmospheric lifetime of 5.9 years, an ozone-depleting potential that is 2.2% of CFC-11 and a global warming potential on a 100-year timescale (GWP-100) of 596 (Burkholder and Hodnebrog, 2023). Simmonds et al. (2017) showed that global emissions of HCFC-124 derived from atmospheric mole fraction measurements decreased monotonically from 2004 to 2015. Earlier estimates using inventory-based methods estimated that emissions of HCFC-124 grew from 0 to 3.3 Gg yr$^{-1}$ between 1990 and 2001 (Ashford et al., 2004). Laube and Tegtmeier (2023) reported that the global mole fraction of HCFC-124 fell from 1.1 ppt in 2016 to 0.9 ppt in 2020. HCFC-124 emissions in China were reported to be around 0.8 Gg yr$^{-1}$ between 2011-2017 (Fang et al., 2019). It has been used as a blowing agent for open-cell foams, as a specialist refrigerant, in very minor quantities as a fire extinguishing agent and in sterilant mixtures (HTOC, 2014;

McCulloch and Midgley, 1998; TEAP, 2024). HCFC-123 has an atmospheric lifetime of around 1.3 years, an ozone-depleting potential that is 2% of CFC-11 and a GWP-100 of 91 (Burkholder and Hodnebrog, 2023). Its atmospheric abundance was measured during a campaign from 2014 to 2016 in Taiwan, but the measurements were not calibrated (to give a reliable mole fraction) and showed no obvious trend (Adcock et al., 2018). Earlier measurements were made 1998-2004 at Cape Grim (now
Kennaook/Cape Grim) in Tasmania, Australia, and increased from 0.05 ppt in January 1998 to 0.13 ppt in August 2004 with variability due to seasonal losses between these dates (Krummel et al., 2006). There is uncertainty in the calibration scale traceability in these measurements and we have decided not to include these in this study. Emissions of HCFC-123 have not previously been quantified using measurements. An inventory method published in 2004 projected that emissions could reach 5.6 Gg yr$^{-1}$ by 2015, using now outdated policy and reduction measures (Ashford et al., 2004). Emissions were estimated to
be around 0.13 Gg yr$^{-1}$ in 2006 using a different inventory method (Wuebbles and Patten, 2009). The main dispersive use of HCFC-123 was in low pressure air conditioning as a replacement for CFC-11, in the HCFC Blend B fire-extinguishing agent, and in very minor uses as a foam blowing agent (FSTOC, 2022; McCulloch and Midgley, 1998; Wuebbles and Patten, 2009). The global bank of chillers contained an estimated 13-15 Gg of HCFC-123 in 2006 (Wuebbles and Patten, 2009).

Both HCFC-123 and HCFC-124 are known to at least partly degrade in the atmosphere to trifluoroacetic acid (TFA) follow-
ing their reaction with hydroxyl radicals (Ellis et al., 2001). TFA has an extremely long lifetime, with no known significant natural loss process, and accumulates in water bodies in the natural environment. TFA is thought to be mildly phytotoxic at high concentrations (Berends et al., 1999). Additionally, HCFC-123 and HCFC-124 are considered per- and polyfluoroalkyl substances, commonly called PFAS, under the 2021 OECD definition (Dalmijn et al., 2024; Wang et al., 2021).

HCFC-123 and HCFC-124 are intermediates during the production of HFC-125 ($C_2HF_5$) (MCTOC, 2022), where per-
chloroethylene ($CCl_2CCl_2$) is reacted with hydrogen fluoride, HF, to substitute its chlorine atom for a fluorine atom to make HCFC-123, HCFC-124 and then HFC-125. Over 90% of HFC-125 is produced using this chemical pathway (TEAP, 2024), following the reactions

$$CCl_2CCl_2 + HF \longrightarrow C_2HCl_4F$$
$$HCFC\text{-}121$$
$$C_2HCl_4F + HF \longrightarrow C_2HCl_3F_2 + HCl$$
$$HCFC\text{-}122$$
$$C_2HCl_3F_2 + HF \longrightarrow C_2HCl_2F_3 + HCl$$
$$HCFC\text{-}123$$
$$C_2HCl_2F_3 + HF \longrightarrow C_2HClF_4 + HCl$$
$$HCFC\text{-}124$$
$$C_2HClF_4 + HF \longrightarrow C_2HF_5 + HCl$$
$$HFC\text{-}125.$$

It was estimated that 10-100 Gg of HCFC-124 was produced in 2020 for use as feedstock (TEAP, 2024). The isomer of HCFC-124, HCFC-124a ($CHF_2CClF_2$), is an intermediate in one production route for HFC-134a ($CH_2FCF_3$), involving CFCs in the production process (Daniel and Reimann, 2023; MCTOC, 2018), but we do not discuss the isomer HCFC-124a in this work as its measurement in the atmosphere is not available. HCFC-123 can be used in the production of CFC-113a ($CCl_3CF_3$), which has multiple uses as a chemical feedstock, including the production of HFO-1336mzz ($CF_3CHCHCF_3$) (MCTOC, 2022). Additional feedstock uses of HCFC-123 are to produce fluorolacton, gamma-cyhalothrin, trifluoroacetyl chloride, trifluoroacetic acid, and pharmaceutical and agricultural products (Miller and Batchelor, 2012). It has been reported that HCFC-123 feedstock produced in China has been exported to Australia to produce medical products (MLF, 2024). Less than 10 Gg yr$^{-1}$ were estimated to have been produced as a feedstock in 2021 (TEAP, 2024). The isomer HCFC-123a ($CHClFCClF_2$) can be produced during the abiotic degradation of CFC-113 (Archbold et al., 2012), although we deem that this source is negligible given the small expected bank size and low levels of landfilling of CFC-113 (Lickley et al., 2022). HCFC-123a is not currently measured in the atmosphere, and we do not discuss it further in this work.

Here we present atmospheric records for HCFC-123 and HCFC-124, since 1993 and 1978, with higher frequency measurements used since 2017 and 1998, respectively. We mainly focus on HCFC-124 given its longer higher frequency measurement record and wider spatial measurement coverage. Using these records, we test the hypothesis of whether it is feasible that a global increase in emissions of HCFC-124 could be from its use during HFC-125 production alone, rather than from a change in emissions from the bank of HCFC-124. Section 2 describes the methods used to quantify emissions of HCFC-123 and HCFC-124, both globally and in Europe, East Asia and the USA, and to determine the contribution to global emissions of HCFC-124 leakage during HFC-125 production. Section 3 presents the results of these analyses, which are discussed in Section 4. Concluding remarks are made in Section 5.

## 2 Methods

### 2.1 Measurements of HCFC-124 and HCFC-123

HCFC-124 was measured as dry-air mole fractions by the Advanced Global Atmospheric Gases Experiment (AGAGE) in situ measurement network (Prinn et al., 2018) and the National Oceanic and Atmospheric Administration (NOAA) Global Monitoring Laboratory's discrete flask measurement network. Measurements of HCFC-123 are only from the NOAA network. Details on the measurement sites used from each network for global, East Asian and European emissions estimates are in Table S1, and in the Supplementary Information for the measurement sites to quantify emissions for the USA.

Regular measurements of HCFC-124 from the AGAGE network began in 1998 using a gas chromatography mass spectrometry (GC-MS) adsorption–desorption system (ADS) (Prinn et al., 2000; Simmonds et al., 1995). We use measurements from the Mace Head, Ireland, and Kennaook/Cape Grim, Tasmania, Australia sites in this work (see Table S1 for details). Measurements, starting in 2003 until present, were made using a Medusa GC-MS instrument at all measurement sites (Arnold et al., 2012; Miller et al., 2008). Further details of the AGAGE instrumentation and calibration used to measure HCFC-124, and the associated errors, can be found in Simmonds et al. (2017).

The NOAA measurements of HFC-124 and HCFC-123 are made by as part of the NOAA Global Greenhouse Gas Reference Network (GGGRN). Samples are collected approximately weekly at each measurement site. Sample collection consists of pressurisation of paired 3 litre electropolished stainless steel flasks to approximately 3 bar absolute. The flasks are returned to NOAA's Global Monitoring Laboratory in Boulder, Colorado, USA for analysis on the Perseus GC-MS instrument. Measurements are made by extracting 480 ml (STP) of air from each flask twice, for a total of 4 injections per sampling event. Reported values are the mean of these 4 measurements. The measurements of HCFC-123 and HCFC-124 are made on the NOAA-2018 and UB-98-b calibration scales, respectively. The UB-98-b scale is derived from the UB-98 scale used by AGAGE. Measurement errors are approximately 3% for HCFC-123 and 2% for HCFC-124.

To extend the measurement record from both networks farther back in time, we supplement the regular measurements of HCFC-123 and HCFC-124 with measurements of archived air (air that had been collected in the past in pressurised air cylinders and measured later). For measurements of HCFC-123 and HCFC-124 from the NOAA network, the archived air measurements were made on the same Perseus GC-MS instrument as the regular measurements. This archive contains 13 measurements of air collected at Niwot Ridge, Colorado, USA, at various dates between 1993 and 2015 and a single measurement of air collected at CGO in 2017. The measurement errors of these archived air samples are around 9% for HCFC-123 and 5% for HCFC-124, which is larger than that of the regular measurement network. Archived air samples measured by the AGAGE network were first collected in 1978 at Kennaook/Cape Grim as part of the Cape Grim Air Archive (Fraser et al., 1991; Langenfelds et al., 1996), and were later analysed on the Medusa GC-MS system (Ivy et al., 2012; Miller et al., 2010; Vollmer et al., 2016). These southern hemispheric air measurements were supplemented by additional measurements of northern hemispheric air collected at Trinidad Head and Scripps Institution of Oceanography, La Jolla, California, USA. The uncertainties in the measurement of HCFC-124 from archived air samples within the AGAGE network are typically around 4%.

## 2.2 Derived global mole fractions and emissions

Global and semi-hemispheric mole fractions are derived concurrently with emissions for HCFC-123 and HCFC-124. These estimates use the measurements made at the locations described in Table S1 and a 12-box model of atmospheric transport (Cunnold et al., 1983; Rigby et al., 2014, 2013). This approach has been detailed previously for HCFC-124 (Simmonds et al., 2017) and is an update to Laube and Tegtmeier (2023), except that the prior mean global emissions are taken from Scenario 2 in Ashford et al. (2004), repeating the 2015 values into the future.

For HCFC-123, the approach follows that of HCFC-124. The prior mean global emissions for HCFC-123 are also taken from Scenario 2 in Ashford et al. (2004), repeating the 2015 values into the future. The stratospheric lifetime is assumed to be 25.7 years (Burkholder and Hodnebrog, 2023), and its reaction rate with OH is taken from Burkholder et al. (2020). HCFC-123 is assumed to have a lifetime with respect to loss to the ocean of 355 years (Yvon-Lewis and Butler, 2002). Uncertainty in the total lifetime is assumed to be 20%. Uncertainty in the measurement calibration scale is unknown, and therefore we have conservatively assumed this to be $\pm 20\%$.

## 2.3 Regional emissions estimates

Emissions of HCFC-124 are derived and presented for western Europe, East Asia and the United States of America (USA). We define western Europe as Belgium, Germany, France, Ireland, Luxembourg, the Netherlands and the United Kingdom. East Asia is defined as eastern China (comprising of the provinces Anhui, Beijing, Hebei, Jiangsu, Liaoning, Shandong, Shanghai, Tianjin and Zhejiang), Japan, North Korea and South Korea. The USA comprises of the contiguous USA (i.e., excluding the states of Hawaii and Alaska, and overseas territories). Emissions in Europe are derived using Zeppelin, Mace Head, Tacolneston, Taunus, Jungfraujoch and Monte Cimone measurements, and emissions in East Asia are derived using Gosan measurements. See Supplementary Table 1 for more information on these measurement stations. Emissions from the USA use measurements from 35 measurement sites, and information on these sites is contained in the Supplementary Information. No emissions for HCFC-123 are derived in these regions due to insufficient measurement coverage.

Emissions in both regions were derived using two inverse systems, the Regional Hierarchical Inverse Modelling Environment (RHIME) (Ganesan et al., 2014; Say et al., 2020; Western et al., 2021; Saboya et al., 2024) and the Inversion Technique for Emission Modelling (InTEM) (Arnold et al., 2018; Manning et al., 2021; Redington et al., 2023). Both use the backward-running Lagrangian atmospheric particle dispersion model NAME from the UK Met Office (Jones et al., 2006) to derive the sensitivity of measurements to emissions. Details of the configurations of the NAME simulations, including the computational domain and treatment of boundary conditions can be found in previous literature for both RHIME (Saboya et al., 2024; Western et al., 2021) and InTEM (Manning et al., 2021). InTEM and RHIME averaged the measurement data into 4 hourly bins for the inversion.

For East Asia, both inverse systems used a priori emissions of $1.0$ Gg yr$^{-1}$ in eastern China, $0.06$ Gg yr$^{-1}$ in South Korea, $0.08$ Gg yr$^{-1}$ in North Korea and $0.2$ Gg yr$^{-1}$ in Japan. These emissions were assumed to be spatially uniform and approximately half of the global emissions originate from all of East Asia. InTEM assumes that the prior uncertainty is normally distributed with the a priori flux as the mean and an uncertainty of 1000% . RHIME assumes a prior log-normal probability distribution with a mean of 1 and a standard deviation of 8, corresponding to $\sim \mathcal{LN}\left(\mu = -2.087, \sigma = 2.043\right)$. In this region, NAME was driven by meteorological output from the global configuration of the Met Office Unified Model (Bush et al., 2023).

For Europe, InTEM used prior emissions of $0.2 \pm 1.3$ Gg yr$^{-1}$ over western Europe with a uniform spatial flux distribution. RHIME used a priori emissions of $0.06$ Gg yr$^{-1}$, with a log-normal uncertainty with the same shape parameters as for East Asia. Given that the bank size and expected emissions of HCFC-124 in Europe is not known a priori, these choices of prior mean emissions are fairly arbitrary, but with large enough uncertainties that they should have little influence on the posterior emissions. Over the United Kingdom and Ireland, NAME was driven by meteorology from the UK-V (high resolution, $\sim$1.5 km) output from the Met Office Unified Model (Tang et al., 2013) embedded within the global meteorology.

The a priori mole fractions at the boundaries of the inversion domain in RHIME are taken from the semi-hemispheric mole fractions derived in Section 2.2. A prior truncated normal probability distribution with a lower bound at 0, a normal-equivalent mean of 1, and a standard deviation of 0.1 is placed on a scaling of the a priori boundary conditions. A single value per month is used to infer the boundary mole fractions at each edge of the domain. Treatment of the baseline and how it is solved for in

InTEM is described in Manning et al. (2021). InTEM additionally solves for potential bias between the European measurement stations. The propagation and management of all aspects of uncertainty in InTEM is described in Manning et al. (2021).

In these inversions, RHIME applied a uniform prior distribution ranging from 0.1 to 1.0 ppt as the model uncertainty, which is treated as a hyperparameter (see Ganesan et al., 2014). This prior range is only weakly informative, allowing the inversion to inform this model uncertainty insofar as possible. The model-data uncertainty is estimated by summing the measurement uncertainty and the model uncertainty in quadrature. A minimum value for the model-data uncertainty was determined based on the annual average difference between the monthly median and the fifth percentile mole fraction. The RHIME inversion domain is partitioned into 250 spatial basis functions, the size of which was determined using a weighted algorithm (see Saboya et al., 2024), and the boundary conditions were estimated for each of the four cardinal directions.

Emissions of HCFC-124 for the USA were derived for the period 2016–2023 using the inversion system developed by NOAA, initially described by Hu et al. (2017, 2016, 2015). The a priori estimate of emissions 0.42 Gg yr$^{-1}$, spatially allocated at $1° \times 1°$ resolution using population density as a scaling factor (giving an equal flux per unit area to western Europe). Prior uncertainties were estimated using a maximum likelihood approach (Michalak et al., 2005; Hu et al., 2015). Sensitivity footprints were generated using the Hybrid Single-Particle Lagrangian Integrated Trajectory (HYSPLIT) model driven by the 12-km North American Mesoscale Forecast System nested with the Global Forecast System . See Hu et al. (2017, 2016, 2015) for more details on the set up of these footprints. Posterior emissions and their uncertainties were derived from two inversion runs with considerations of one prior emissions and two background estimation techniques (see, e.g., Western et al., 2022).

## 2.4 Source sector estimation of HCFC-124 emissions

We follow the methodology of (Bourguet and Lickley, 2024) to decouple emissions of HCFC-124 from its dispersive uses and emissions from HFC production. Briefly, the methodology builds upon a Bayesian framework originally developed by Poole and Raftery (2000) and applied in Lickley et al. (2020) and Lickley et al. (2021). This method uses prior knowledge of the size of emission sources and emissions factors of HCFC-124, along with the measurements of the atmospheric mole fraction of HCFC-124 derived in Section 2.2 (using only AGAGE measurements, given the longer higher-frequency measurement record, see Figure 1), to derive an estimate of emissions of HCFC-124 from HFC-125 decoupled from other sources. This estimate relies on the assumption that the use of HCFC-124 as an intermediate for HFC-125 production is the only unreported source of HCFC-124 production.

### 2.4.1 Simulation model

The calculation of the likelihood of the parameters of interest relies on a forward simulation of global mean mole fraction to transform these parameters into an observable quantity, which in this case is the global mean mole fraction. We simulate the global mean mole fraction following a one-box approach,

$$M_{t+1} = M_t \exp\left(\frac{1}{\tau}\right) + AE_t, \tag{1}$$

where $M_t$ is the global mean mole fraction of HCFC-124 in year $t$, $\tau$ is the total atmospheric lifetime of HCFC-124, $A$ is a factor that converts emissions into global mole fractions (24.26 ppt Gg$^{-1}$), and $E_t$ is the mass of HCFC-124 emitted in year $t$. The simulation was initialised with the observed global mole fractions of HCFC-124 in 2005 and assumes a 1-year mixing timescale in the troposphere.

HCFC-124 emissions in year $t$ were calculated as

$$E_t = I \cdot \text{HFC-125}_t + \sum_i (P_{i,t}UD_i + B_{i,t}F_i), \tag{2}$$

where $P_{i,t}$ is the mass of reported HCFC-124 production for dispersive end-use type $i$ in year $t$, $U$ is an underreporting factor, $D_i$ is the fraction of HCFC-124 emitted during the year of production for dispersive end-use type $i$, $F_i$ is the fraction of HCFC-124 emitted from bank type $i$ in year $t$ (and is constant between years), $\text{HFC-125}_t$ is the mass of HFC-125 produced in year $t$, and $I$ is the leakage rate of HCFC-124 from use as an intermediate in HFC-125 production (relative to HFC-125 production). $B_{i,t}$ is the mass of HCFC-124 stored in bank type $i$ in year $t$ and was calculated as

$$B_{i,t} = (1 - D_i)P_{i,t} + (1 - F_i)B_{i,t-1}, \tag{3}$$

which was first initialised with $B_{i,1992} = 0$ in 1992 (the first year that the production of HCFC-124 was reported) and integrated to 2004. As the observed global mole fraction of HCFC-124 in 2005 cannot be explained by reported production up to that year, we assumed that underreporting occurred through at least 2004. Thus, we set the mass of HCFC-124 stored in banks in 2004 ($B_{i,0}$) as an uncertain parameter with bounds of 100–300% of the reported banked mass. We then calculated bank sizes and emissions from 2005 to 2023 and integrated mole fractions from 2006 to 2024 with this increased bank size.

### 2.4.2 Production data

Prior knowledge of the production of HCFC-124, $P_{i,t}$, using data reported to the United Nations Environment Programme is used to inform the estimation of the parameters of interest. The production data reported to UNEP are the total production for all end-uses and, separately, as production for feedstock uses. To avoid double counting, we calculated the dispersive production by subtracting the feedstock production from the total production. The data do not contain information on how the feedstock production was used. We assume that all reported HCFC-124 feedstock production was used in HFC-125 production and therefore is captured by $\text{HFC-125}_t$. Reported production in 2003 and 2004 was anomalously low (around two orders of magnitude smaller than in previous and following years) and thus we chose to begin our simulations in 2005, and production data are used from 2005-2023.

For emissions from use as an intermediate in HFC-125 production, we used a previously published estimate of HFC-125 production (Velders et al., 2022). The production estimates are constrained by observed mole fractions through 2019 and are scaled based on population growth while adhering to the Kigali Amendment from 2020 to 2023. As the chemical conversion rate of HCFC-124 to HFC-125 production is not known, we used this HFC-125 production estimate directly in our emissions calculation and employed an emission rate ($I$) that is relative to HFC-125 production (Eq. 2). We do not assign an uncertainty to HFC-125 production, as this would be linearly compensated by $I$.

### 2.4.3 Prior parameter distributions

Unless otherwise stated, the prior distributions for the parameters of interest were assumed to follow a beta distribution, $\text{Beta}(2,2)$. All parameters are assumed to be constant in time, meaning that any potential variability in the leakage rates from banks or emission rates during production or use as an intermediate are not captured by this model.

We assumed a normal distribution for the inverse atmospheric lifetime of HCFC-124, with a mean of $\frac{1}{5.9}$ yr$^{-1}$ and 20% 1-sigma uncertainty. As underreporting of ODSs has been documented previously (e.g., Montzka et al., 2018), the underreporting factor $U$ was assumed to be a beta distributed between 0.8 and 1.75 with probability $\text{Beta}(2,5)$. The allocation of the mass of reported HCFC-124 production, $P_{i,t}$, across bank types $(i)$ was informed by production reported to the Alternative Fluorocarbons Environmental Acceptability Study (AFEAS) from 1990–2003. We assumed that 1% of dispersive production was stored in long-lived banks, 20–40% of dispersive production was stored in short-lived banks, and the remainder was stored in medium-lived banks. The fraction of HCFC-124 emitted during production and from the bank, $D_i$ and $F_i$, were also informed by AFEAS reporting. The prior distributions follow a $\text{Beta}(2,2)$ distribution between the ranges, where the subscripts $s$, $m$ and $l$ correspond to the short, medium and long-lived banks, respectively; $D_s$, 70%–90%; $D_m$, 20–40%; $D_l$, 0–5%; $F_s$, 100%; $F_m$, 12.5–32.5%; and $F_l$, 1–3%. The initial bank size, $B_{i,0}$, was assumed to be between 100–300% of the bank size that would be implied by reported HCFC-124 production from 1992–2004, following a uniform distribution between these upper and lower values. The leakage rate of HCFC-124 from its use as an intermediate in HFC-125 production, $I$, was informed by an estimated feedstock emission rate of 1.5–6.1% (MCTOC, 2022). However, after simulating mole fractions with this distribution, $I$ was adjusted to 0.5–3.5% to decrease the parameter space with a conditional probability near zero.

### 2.4.4 Posterior estimation

We estimated the posterior distributions using the sampling importance resampling method (Bates et al., 2003; Hong et al., 2005; Rubin, 1988). To implement this, we independently sampled each prior distribution 1,000,000 times and simulated a mole fraction time series with each combination of parameter values. Posterior distributions were then estimated using a likelihood function that is a function of the difference between modelled and observed mole fractions. As simulated mole fractions depend on emissions in the year prior, all parameters are constrained through 2023. For more details on this method, see Bourguet and Lickley (2024).

## 3 Results

### 3.1 Atmospheric abundance and global emissions of HCFC-124 and HCFC-123

Records of atmospheric HCFC-124 mole fractions based on in situ measurements from the AGAGE network are available from 1987 through 2023 and are shown in Figure 1 (from 1995 onwards). Regular measurements from the NOAA network have been available since 2015, and infrequent measurements from archived air from the NOAA network have been available since 1993. The global mean mole fraction of HCFC-124 has fallen year-on-year since its peak of $1.48 \pm 0.03$ ppt in 2007, declining to 0.85

± 0.02 ppt in 2023 using AGAGE measurements. NOAA measurements show a peak in 2009, although regular sampling was not performed at this time, and declined to 0.85 ± 0.02 ppt in 2023. The rate of decline slowed around 2019, consistent with an increase in the mole fraction in the Northern Hemisphere. Persistent higher mole fractions in the Northern Hemisphere are indicative of continued higher emissions in the Northern Hemisphere compared to the Southern Hemisphere. These results are supported by NOAA sampling. Estimated emissions of HCFC-124 using AGAGE measurements have also shown a persistent decline from 7.1 ± 1.1 Gg yr$^{-1}$ (± 1 s.d.) in 2003 to 2.5 ± 0.8 Gg yr$^{-1}$ in 2019, after which emissions increased again, reaching 3.3 ± 0.7 Gg yr$^{-1}$ in 2023. Using NOAA measurements, estimated emissions fell to their lowest value in 2018 (2.4 ± 0.9 Gg yr$^{-1}$) and rose to 2.8 ± 0.7 Gg yr$^{-1}$ in 2023. Figure 3 shows the derived emissions.

The record of atmospheric HCFC-123 mole fractions using regular NOAA flask sampling is available only from 2017 through 2023, with sporadic archived samples extending the record back to 1993 as shown in Figure 2. During the period 2017-2023, the global mean mole fraction of HCFC-123 was near-unchanged at around 0.23-0.25 ppt. The lack of decline in the atmospheric abundance is reflected in the global emissions, shown in Figure 3, which have steadily increased over the record to 5.0 ± 1.0 Gg yr$^{-1}$ in 2023.

### 3.1.1 Emissions of HCFC-124 during HFC-125 production

Using the approach described in Section 2.4, we estimate that global emissions of HCFC-124 from HFC-125 production increased from 0.5 (0.5-0.6) Gg yr$^{-1}$ to 2.8 (2.4-3.1) Gg yr$^{-1}$ (mean and 68% uncertainty) between 2005 and 2023. The trend is opposite for emissions from banks, which decreased from 5.6 (5.0-6.2) Gg yr$^{-1}$ to 0.4 (0.3-0.5) Gg yr$^{-1}$ between 2005 and 2023. Using this methodology, the total emissions of HCFC-124 decreased from 6.1 (5.4-6.8) to 3.2 (2.8-3.6) Gg yr$^{-1}$ over the same period, similar to those derived from top-down emissions using AGAGE measurements, which decreased from 6.3 ± 1.2 Gg yr$^{-1}$ to 3.3 ± 0.7 Gg yr$^{-1}$ between 2005 to 2023. Figure 4 shows the comparison of these emissions, and their sum, to those derived in Section 3.1.

### 3.2 Emissions of HCFC-124 from western Europe

Emissions of HCFC-124 from western Europe (see supplementary information for data values) are presented in Figure 5a for the two inverse models described in the Methods, and the spatial distribution of these emissions are shown in in Figure 5b using the mean of the two inverse models over 2008-2023. Emissions show a continuous decrease since 2008 and are much smaller than the global emissions, representing around 1%, or 25 tonnes yr$^{-1}$, since 2019. The mean spatial distribution of emissions in Europe (Figure 5b) shows no obvious location where emissions are substantially and persistently larger than elsewhere in the region. Emissions of HCFC-124 appear to have mostly declined in line with its production phase-out in Europe.

### 3.3 Emissions of HCFC-124 from East Asia

Emissions of HCFC-124 from East Asia are less than 0.1 Gg yr$^{-1}$ in Japan, North Korea and South Korea over 2008-2023 (see supplementary information). Figure 6a shows emissions from eastern China, which have increased from around 0.2 Gg

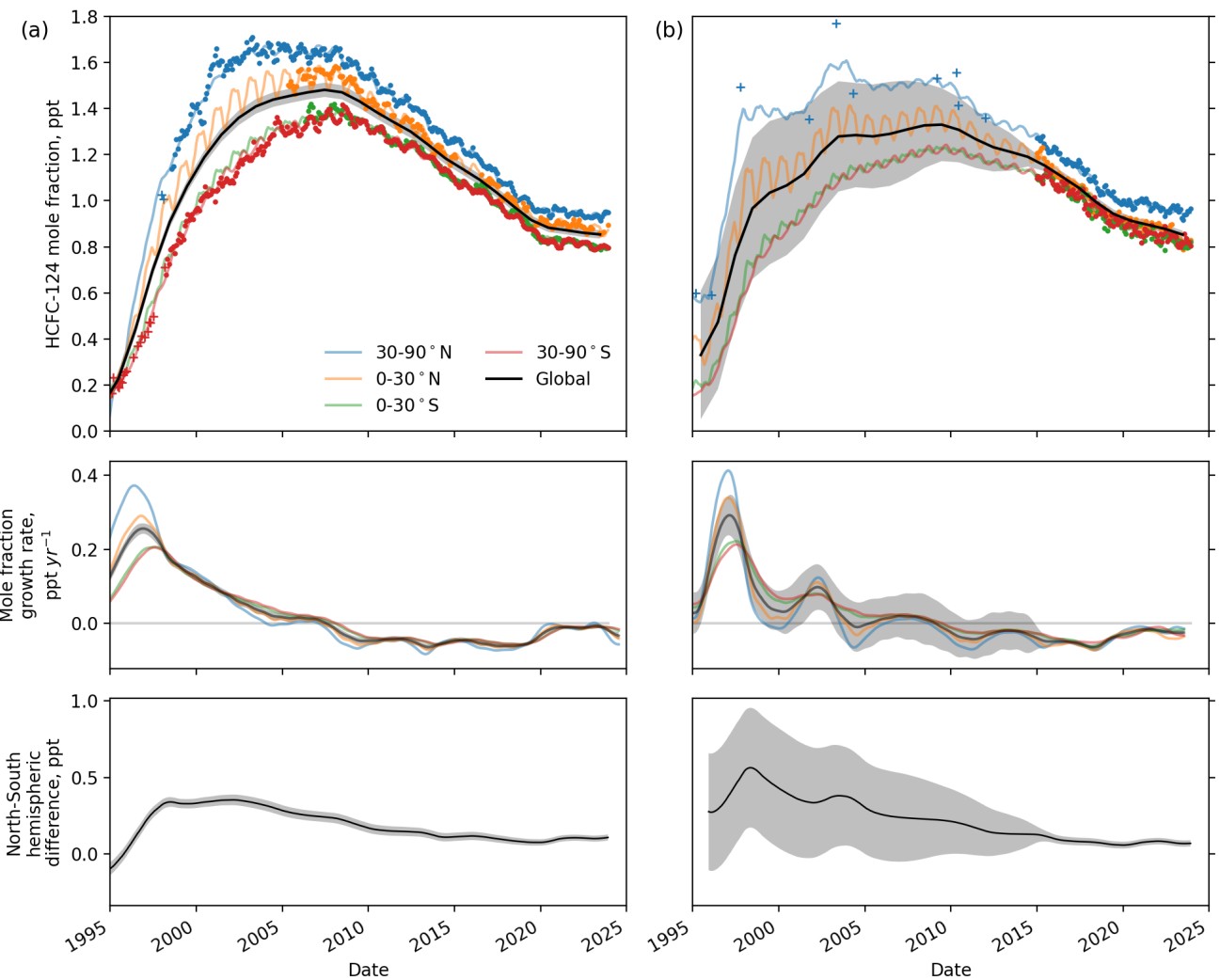

**Figure 1.** Global and semi-hemispheric mole fractions of HCFC-124 derived from measurements from the (a) AGAGE and (b) NOAA networks. The top row shows the global and semi-hemispheric mole fractions, the middle row shows the global and semi-hemispheric mole fraction growth rates, and the bottom row shows the north-south hemispheric differences in the mole fractions. Solid circles show semi-hemispheric averages of HCFC-124 from regular measurements and crosses show measurements of archived air. Grey shading shows the 1 s.d. uncertainties in the global quantities. See Supplementary Table S1 on information on the measurement stations used.

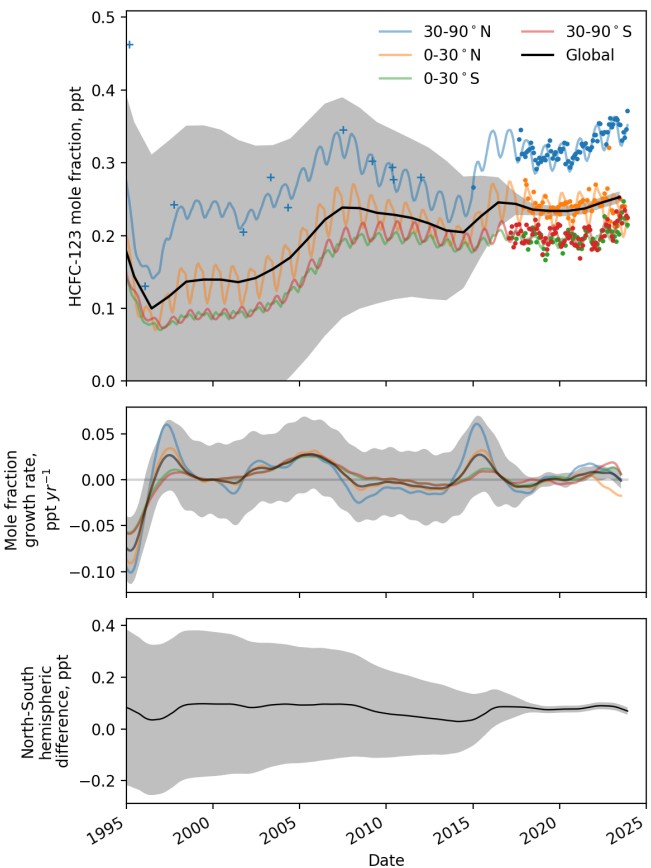

**Figure 2.** Global and semi-hemispheric mole fractions of HCFC-123 from the NOAA network. The top row shows the global and semi-hemispheric mole fractions, the middle row shows the global and semi-hemispheric mole fraction growth rates, and the bottom row shows the north-south hemispheric differences in the mole fractions. Solid circles show semi-hemispheric averages of HCFC-124 from regular measurements and crosses show measurements of archived air. Grey shading shows the 1 s.d. uncertainties in the global quantities. See Supplementary Table S1 on information on the measurement stations used.

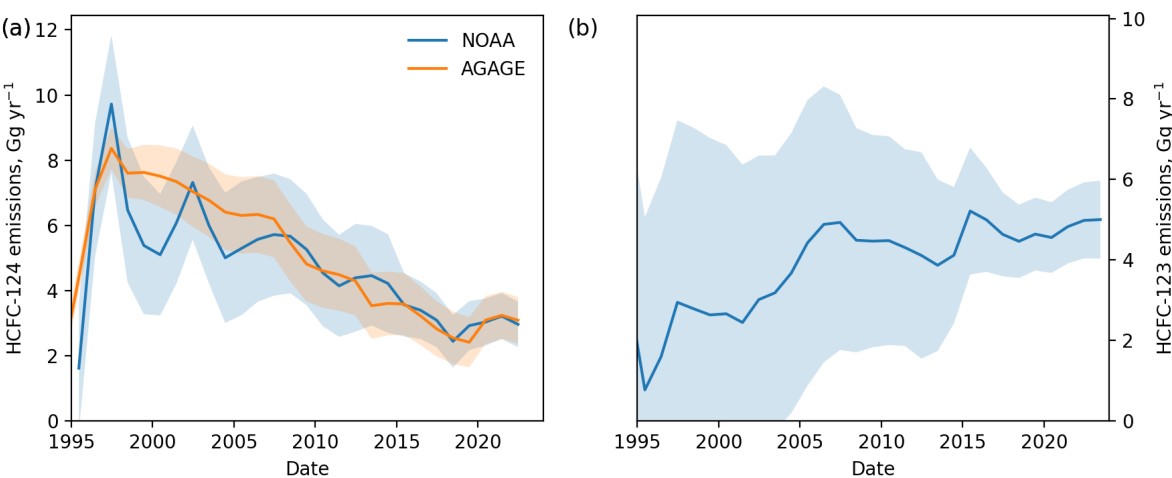

**Figure 3.** Global emissions of (a) HCFC-124 and (b) HCFC-123. Emissions derived from measurements made by the AGAGE network are shown by the orange line and shading and by the NOAA network in blue. Shading shows the 1 s.d. uncertainty in the mean derived emissions.

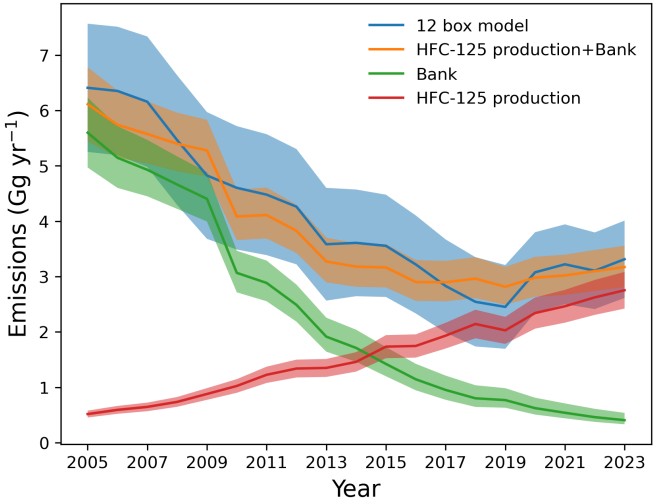

**Figure 4.** Derived global emissions of HCFC-124 separated into the contribution from the bank of reported production (green line and shading) and that estimated to come from HFC-125 production (red line and shading) and the sum of these two sources (orange line and shading). The blue line and shading show the global emissions derived from AGAGE measurements in Section 2.2 and presented in Figure 1 Shading shows 1 s.d. uncertainties.

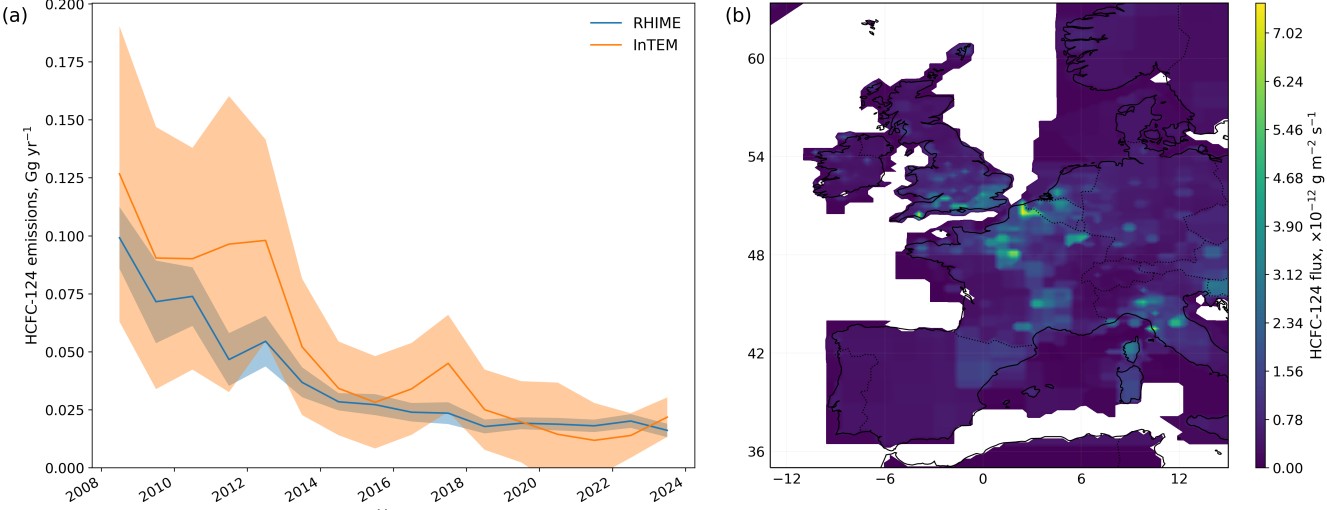

**Figure 5.** (a) Annual emissions of HCFC-124 from western Europe (defined as Belgium, Germany, France, Ireland, Luxembourg, the Netherlands and the United Kingdom) from two inverse methods, RHIME (blue line and shading) and InTEM (orange line and shading). Uncertainties for InTEM are 1 s.d. and for RHIME are the 68% uncertainty interval. (b) The mean spatial distribution of emissions in western Europe over the period 2008-2023 using the mean flux for RHIME and InTEM for each year.

$yr^{-1}$ in 2008 to around 0.45 Gg $yr^{-1}$ in 2023. There is no obvious trend in the emissions in eastern China since around 2015, where the emissions estimate from RHIME and InTEM appear to fluctuate around a value of around 0.45 Gg $yr^{-1}$. Figure 6b shows the mean spatial fluxes from East Asia over 2008-2023, which suggests that emissions are larger in the major industrial

provinces of Shandong, Hebei, Jiangsu and Zhejiang. Emissions are also higher in the Seoul region of South Korea compared to many other areas in East Asia.

### 3.4   Emissions of HCFC-124 from the United States of America

Emissions of HCFC-124 from the USA are available for the period 2016-2023. Over this period, emissions decreased from 0.55 $\pm$ 0.03 Gg $yr^{-1}$ in 2015 to 0.26 $\pm$ 0.04 Gg $yr^{-1}$ 2023, shown in Figure 7a. The mean spatial distribution of these emissions is

shown in Figure 7b, noting that population-weight a priori emissions were used to derive emissions for the USA (as opposed to no spatial weighting for western Europe and East Asia). There are no obvious locations of large emissions, except for the population centres imposed by the a priori emissions, which makes interpretation of the spatial distribution very uncertain.

### 4   Discussion

Based on atmospheric measurements from the AGAGE and NOAA networks, global emissions of HCFC-124 have increased

since 2019 despite a phase-down of HCFC production for emissive uses. The production of HFC-125 has been forecast to

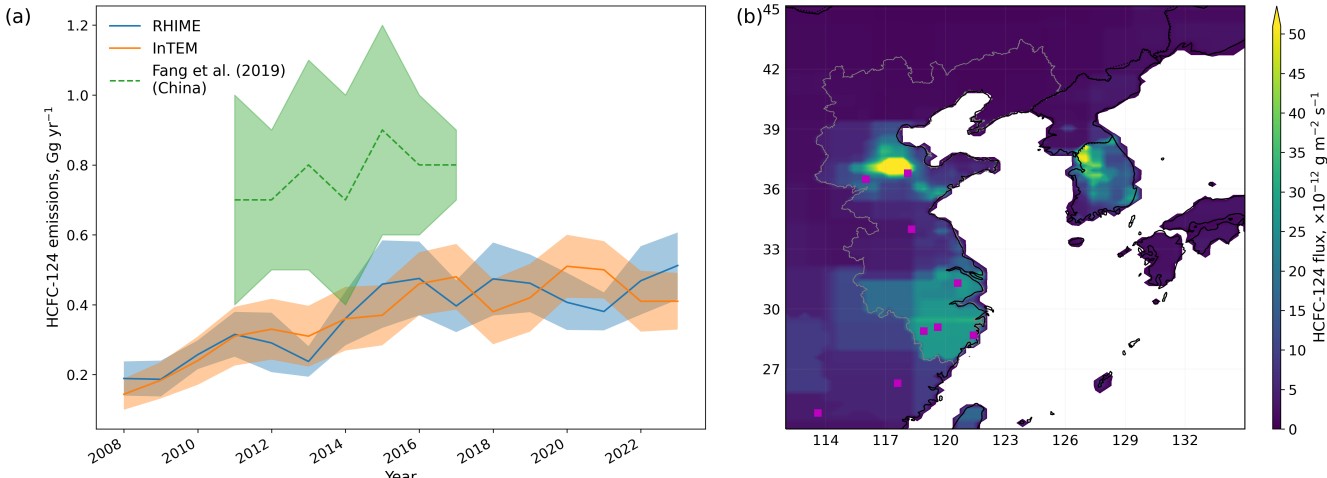

**Figure 6.** (a) Annual emissions of HCFC-124 from eastern China (see main text for regional definition and shown by grey lines in figure (b)) from the RHIME (blue line and shading) and InTEM (orange line and shading) inversion models. Emissions from South Korea, North Korea and Japan are all less than 0.1 Gg yr$^{-1}$ each year. Uncertainties for InTEM are 1 s.d. and for RHIME are the 68% uncertainty interval. The green dashed line and shading shows the emissions previously estimated for the whole of China (Fang et al., 2019). (b) The mean spatial distribution of emissions in East Asia over the period 2008-2023 using the mean flux for RHIME and InTEM for each year. Magenta squares show the locations of known HFC-125 production facilities in China in 2023.

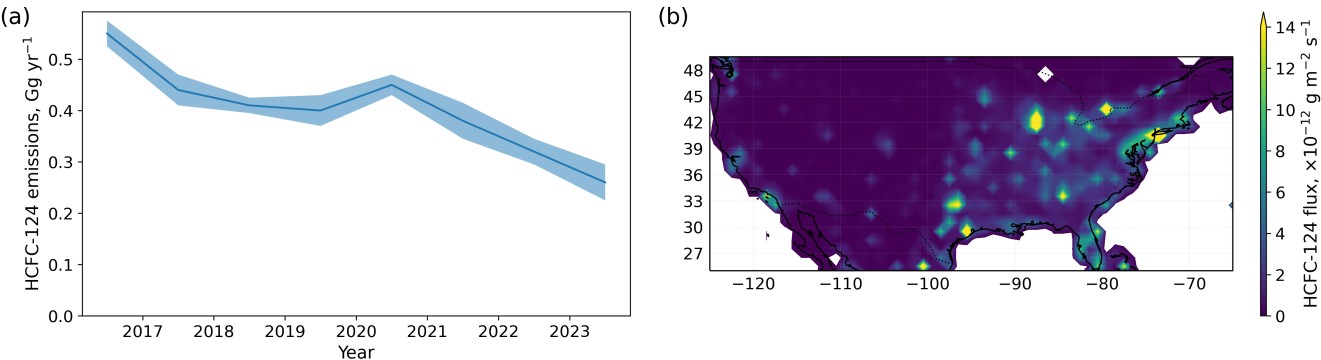

**Figure 7.** (a) Annual emissions of HCFC-124 from the USA (excluding Alaska, Hawaii and overseas territories) 2016-2023. Uncertainties are 1 s.d. and given by the shading. (b) The mean spatial distribution of emissions over the USA for the period 2016-2023. Note that this inversion used an a priori emissions estimate where the emissions were weighted in space by the population. The posterior mean emissions estimates largely follow this spatial pattern and are highly uncertain.

increase until around 2027, assuming universal compliance with the Kigali Amendment (Velders et al., 2022). HCFC-124 production for use in appliances should not be increasing, given the phase-out schedule under the Montreal Protocol, and we know of no reason why the rate of emissions from the bank should suddenly increase. With a lack of another feasible hypothesis as to why emissions of HCFC-124 would increase, we have tested the hypothesis of whether the increase in emissions of

HCFC-124 (and HCFC-123) can be due to a change in the emissions of intermediates and impurities during the production of HFC-125.

We have also presented the first atmospheric record of the global abundance and emissions of HCFC-123. Like HCFC-124, HCFC-123 is an intermediate during HFC-125 production. Its emissions have not declined, and possibly increased, over 2018-2023, in line with an increase in HCFC-124 emissions over 2019-2023. The emissions trend of HCFC-123 since 1993 is

uncertain and does not appear to follow that of HCFC-124, perhaps due to their different dispersive uses prior to their phase-down. There has been a possible increase in the emissions of HCFC-123 since the early 2000s, which is similar to the timing of a reduction in HCFC consumption in developed countries, which was largely replaced by HFC consumption (Velders et al., 2022). It has been reported that HCFC-123 is practically completely consumed during HFC-125 production and that leakage of HCFC-124 is likely to be higher than HCFC-123 as HCFC-124 is involved in the final fluorination step to HFC-125 (TEAP,

2003). If all emissions of HCFC-123 are from HFC-125 production, the emissions of HCFC-123 represent around 2% by weight of HFC-125 production, which is much larger than the previously estimated leakage rate of 0.001–0.01% for HCFC-123 during HFC-125 production (TEAP, 2003). Sources of emissions of HCFC-123 other than during HFC-125 production, including a bank of unknown size, will also contribute to the global emissions. A report projecting HCFC-123 bank emissions from 2002 onwards estimated that emissions from the bank could be between 1.6-5.6 Gg yr$^{-1}$ by the year 2015 (Ashford

et al., 2004), compared to our estimate of $5.1 \pm 1.7$ Gg yr$^{-1}$, noting that the date of the complete phase-out HCFCs under the Montreal Protocol was brought forward by 10 years in 2007. It is not known to what extent HCFC-123 is used to produce CFC-113a and other chemicals. We are unable to disentangle emissions of HCFC-123 from the bank and those from chemical production. As emissions of HCFC-123 in 2023 were larger than those of HCFC-124, and it is expected that less HCFC-123 intermediate will be emitted than HCFC-124 given its earlier fluorination step, it is likely that there is another substantial source

of HCFC-123 in addition to HFC-125 production. Our analysis of HCFC-124 emissions using knowledge of production has allowed separation of emissions from the bank and from chemical-production. Given the short atmospheric record of regular measurements of HCFC-123, and the uncertainty in trends prior to this record, we are unable to draw any firm conclusions, simply observing that the timing of the increase in the global emissions of HCFC-123 and HCFC-124 is similar and that both are intermediates in HFC-125 production.

The global mean emissions of both HCFC-123 and HCFC-124 have large interannual variability between some years. For HCFC-124, the increase after 2019 is of similar magnitude to the interannual variability in some years, particularly using the NOAA record. This may be due to random errors in the estimation procedure or changes in the atmospheric sink and other dynamic effects not considered in the global box model. For HCFC-124, we believe that the increase in emissions after 2019 is genuine, given that the increase in emissions is driven by a slowing in the decline of the background atmospheric abundances,

which are measured directly and precisely, and the global mean is subject to less uncertainty than the derived global emissions.

The result is also robust between the independent NOAA and AGAGE networks in both the abundance trends and emissions. If this impact was due to dynamical or loss related changes, this change would be apparent in other atmospheric compounds with loss process dominated by the hydroxyl radical.

Our analysis of the emissions of HCFC-124 from its use as an intermediate during HFC-125 production assumes that the increase in total global emissions of HCFC-124 is only due to leakage during this process. The derived leakage rate from this process is $1.0 \pm 0.2\%$ by mass of HFC-125 production. The assumed leakage rate during fluorochemical production can differ significantly between literature sources. A report detailing the production of HFC-125, using HCFC-124 as an intermediate, estimated that its leakage rate to be 0.01-0.1% by mass of HFC-125 production (unknown uncertainty range) (TEAP, 2023). Additional reporting suggests a most likely emissions factor of the feedstock itself of 3.6 (1.5-6.1)% (unknown uncertainty range), or 3.1 (1.2-4.9)% if not including any losses during shipping and storage (TEAP, 2024). Other reported estimates have been as high as 4% for fluorochemical production (Calvo Buendia et al., 2019), with a 95% uncertainty range of 0.1–20.0%.

Emissions of HCFC-124 from western Europe were around 20 tonnes per year in 2023 and have monotonically fallen since 2008. Western Europe can therefore be eliminated as a potential region from where emissions of HCFC-124 are increasing. Similarly, emissions from the USA have approximately halved between 2016 and 2023, and were over ten times as large as western Europe in 2023, at around 260 tonnes. This estimate agrees with the bottom-up estimate by the US Environmental Protection Agency, who report that less than 500 tonnes of HCFC-124 (no exact value given) were emitted annually 2018-2022 (EPA, 2024). Conversely, emissions increased in eastern China, from around 0.2 Gg $yr^{-1}$ in 2008 to around 0.5 Gg yr-1 in 2015 and remained near-constant between 2015 and 2023.

China is the world's largest fluorochemical producer, and some production-related emissions of HCFC-124 in China would be expected during HFC-125 production. Emissions from eastern China are currently the largest of the regions studied and is therefore the largest known source region. Figure 6 shows locations of known factories producing HFC-125 in China in 2024 (taken from Ministry of Ecology and Environment of the People's Republic of China, 2024, see Supplement for English translation). Higher emissions in eastern China are generally co-located with the locations of factories producing HFC-125, which supports our suggestion of HFC-125 as a major source of HCFC-124 emissions. However, these regions are highly industrialised, densely populated and contain many fluorochemical production facilities, meaning that this collocation does not preclude other sources of HCFC-124. Emissions from Japan, North Korea and South Korea are not increasing and are less than 0.1 Gg $yr^{-1}$. Emissions of HCFC-124 in the region of Seoul in South Korea are higher than in surrounding areas. The nature of this source is unclear, and South Korea remains a small emitter of HCFC-124. Our analysis of the global increase in emissions of HCFC-124 from HFC-125 production suggests that emissions of HCFC-124 from HFC-125 production had continually increased from 2005-2023 (Figure 4). Therefore, based on the emissions estimates in this work, emissions from East Asia are insufficient to explain the entire global increase in emissions prior to 2015, and cannot explain the increase since 2015. A previous study by Fang et al. (2019), covering the whole of China and using different measurement stations than this work, reported an increase in emissions of HCFC-124 from $0.7 \pm 0.3$ Gg $yr^{-1}$ in 2011 to $0.9 \pm 0.3$ Gg $yr^{-1}$ in 2015, which then remained constant at $0.8 \pm 0.2$ Gg $yr^{-1}$ in 2016 and $0.8 \pm 0.1$ Gg $yr^{-1}$ in 2017. An increase of around 0.2 Gg $yr^{-1}$ between 2011 and 2015 is in line with our derived emissions. The emissions derived by Fang et al. (2019) imply that around

half of the emissions in China are from outside of the region we term eastern China, assuming that there are not systematic biases between the inverse modelling approaches. It remains uncertain which regions are responsible for the global HCFC-124 emissions growth toward the end of our study period.

Some HFC-125 installed in appliances may still be contaminated with intermediates or by-products that were not properly removed during the production process. Possible contamination should remain detectable if the composition of the installed refrigerant is measured. We do not have direct analyses of installed refrigerant composition; however, there are occasions in which the refrigerant in the air conditioning unit at measurement stations has leaked, and the composition is thus measurable. Using events where air conditioning units containing HFC-125 had leaked at six AGAGE measurement stations and at an additional urban station, we can deduce an approximate level of contamination that was present in those air conditioning units (see Supplementary Information). In these cases, the ratio of HCFC-124 to HFC-125 in the leaked refrigerant ranged was up to $4.6 \times 10^{-2}\%$ by mass. Scaling this to global emissions of HFC-125 would give maximum emissions of around 50 tonnes of HCFC-124 emitted globally in 2023. There were no elevated measurements of HCFC-124 in many cases of air conditioning leakage. It is therefore unlikely, based on these few test cases, that contamination in air conditioning units is a source of the increase in HCFC-124 emissions. However, the levels of contamination will differ depending on the manufacturing source of HFC-125.

Western et al. (2023) suggested that an increase in emissions of multiple CFCs since 2010 may be due to by-products from HFC-125 production being emitted to the atmosphere. Our analysis suggests that emissions of HCFC-124 originating from HFC-125 production has increased continuously since 2010. The increase in emissions of CFCs and HCFC-124 does not provide evidence for a shared source process, other than the coincident timing of their increase. Atmospheric measurements previously made in Taiwan found that pollution events of HCFC-124 were highly correlated with other by products of HFC-125 production (Adcock et al., 2018), which suggests that their sources are approximately co-located.

## 5 Conclusions

Emissions of the hydrochlorofluorocarbons HCFC-123 and HCFC-124 have increased since 2019 despite a phase-down of their production and consumption for dispersive uses under the Montreal Protocol. Any increase in the production of HCFCs should only be for their use to produce other chemicals. Both HCFC-123 and HCFC-124 are intermediates in the production of the non-ozone depleting greenhouse gas HFC-125, which has been used to replaced HCFCs for refrigeration and air conditioning and for fire suppression applications. Our analysis suggests that the increase in global emissions of HCFC-124 can be explained from leakage during the production of HFC-125. We estimate this leakage rate as ~1.0% of HFC-125 production. Other sources of emissions cannot be ruled out. However, emissions of HCFC-123, another intermediate in HFC-125 production, also potentially increased with similar timing to HCFC-124.

It is unclear from where the increase in emissions of HCFC-124 since 2019 are originating, as emissions in western Europe, the USA and East Asia have not increased since 2015, unlike the global emissions. Prior to 2019, emissions from western

Europe, the USA and East Asia are insufficient to explain the global changes over this period. An increase in monitoring stations around the world would help to better locate and understand the emission sources of HCFC-124.

The increase in emissions of HCFC-123 and HCFC-124 is likely to have a negligible impact on stratospheric ozone recovery at present, and little impact on the climate. The production of HFC-125 is projected to increase in the coming years (Velders et al., 2022). However, a general transition to lower GWP refrigerants, such as HFC-32, may mean that the phase-down of HFC-125 is faster than scheduled under the Kigali Amendment. There is no quantitively mandated degree to which production of substances controlled under the Montreal Protocol, or their emission, must be mitigated during the production of other

chemicals (except for HFC-23). It may be that leakage rates of feedstocks, intermediates and by-products are higher than previously thought, suggesting that large quantities of ozone-depleting substances and potent greenhouse gases could be emitted from new fluorochemical production for many years to come.

*Code and data availability.* Measurements from the AGAGE network are available at https://doi.org/10.15485/2476540. The most recent measurements from the NOAA network are available at https://gml.noaa.gov/aftp/data/hats/PERSEUS/ (accessed 10 June 2025). All input

and outputs to the 12-box model, and all outputs from the NOAA, RHIME and InTEM inverse models, and outputs from the bank/feedstock model are available at (Western et al., 2025). The code for the bank/feedstock model is available at https://doi.org/10.5281/zenodo.13890803. Code for the 12 box model and its inverse method are available at (Rigby and Western, 2022a) and (Rigby and Western, 2022b). The RHIME inversion code is available at https://doi.org/10.5281/zenodo.10650595. NAME and InTEM are available for research use and subject to licence; please contact enquiries@metoffice.gov.uk for more details. The FLEXPART model is available from https://www.flexpart.eu (Pisso

et al., 2019).

*Author contributions.* LMW conceived and led the study. LMW led the writing of the manuscript with contributions from SB, MC, HdL, AM, MA, PBK, JM, DR, IV, PJF, ALG, SR, KMS, MR and RGP. LMW performed global modelling and analysis of the data. SB performed analysis of emissions from banks and production. HdL, AJM, LMW and LH performed regional inversions. DR analysed data for leakage from air conditioning units. MA supplied information on HCFC uses and production locations in China. New measurements, including from

archived air, were mainly supplied by MC, IV, PBK and JM. Additional measurements, and their quality control and assurance, were supplied by MKV, PBK, JM, JA, AE, PJF, CMH, CL, MM, SAM, DN, SO'D, SP, SR, PKS, RS, KMS, TW, DY and RFW.

*Competing interests.* The authors declare no competing interests.

*Acknowledgements.* We would like to thank all measurement station personnel, without whom this work would not be possible. We thank the UK Met Office for licencing and use of the NAME model and meteorological data from the Unified Model. The AGAGE Medusa GC–MS

system development, calibrations and measurements at the Scripps Institution of Oceanography, La Jolla and Trinidad Head, CA, USA;

Mace Head, Ireland; Ragged Point, Barbados; Cape Matatula, American Samoa; and Kennaook/Cape Grim, Australia were supported by the NASA Upper Atmospheric Research Program in the United States with grants NNX07AE89G, NNX16AC98G and 80NSSC21K1369 to MIT and NNX07AF09G, NNX07AE87G, NNX16AC96G, NNX16AC97G, 80NSSC21K1210 and 80NSSC21K1201 to SIO (and earlier grants). The Department for Energy Security and Net Zero (DESNZ) in the United Kingdom supported the University of Bristol for operations at Mace Head, Ireland and Tacolneston, United Kingdom (contracts 1028/06/2015, 1537/06/2018 and 5488/11/2021) and through the NASA award to MIT with the subaward to University of Bristol for Mace Head and Barbados (80NSSC21K1369). The National Oceanic and Atmospheric Administration (NOAA) in the United States supported the University of Bristol for operations at Ragged Point, Barbados (contract 1305M319CNRMJ0028) and operations at Cape Matatula, American Samoa. In Australia, the Kennaook/Cape Grim operations were supported by the Commonwealth Scientific and Industrial Research Organization (CSIRO), the Bureau of Meteorology (Australia), the Department of Climate Change, Energy, the Environment and Water (Australia), Refrigerant Reclaim Australia, the Australian Refrigeration Council and through the NASA award to MIT with subaward to CSIRO for Cape Grim (80NSSC21K1369). Measurements at Zeppelin are supported by Norwegian Environment Agency. Measurements at Jungfraujoch are supported by the Swiss National Programs HALCLIM and CLIMGAS-CH (Swiss Federal Office for the Environment, FOEN), by the International Foundation High Altitude Research Stations Jungfraujoch and Gornergrat (HFSJG), and by the European infrastructure projects ICOS and ACTRIS. Observations at Gosan, South Korea, are supported by the National Research Foundation of Korea (NRF) grant funded by the Korean government (Ministry of Science and ICT; no. RS-2023-00229318). This research was supported in part by NOAA cooperative agreement NA22OAR4320151. The statements, findings, conclusions, and recommendations are those of the author(s) and do not necessarily reflect the views of NOAA or the U.S. Department of Commerce. The instrumentation at Monte Cimone was funded by the Italian component of ACTRIS (Aerosol, Clouds and Trace Gases Research Infrastructure), under the Programma Operativo Nazionale Ricerca e Innovazione 2014-2020 PIR01_00015 "PER-ACTRIS-IT".

LMW, MR, ALG were supported by the Natural Environment Research Council (NERC) Investigating HALocarbon impacts on the global Environment project (InHALE, NE/X00452X/1) and MR, ALG, HdL were supported by the Horizon Europe PARIS project.

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
