# Peer review of "Impact of leakage during HFC-125 production on the increase in HCFC-123 and HCFC-124 emissions"

_EGUsphere, 2025_

## Referee Comment (RC1)

*Review of draft paper*

**"Increasing emissions of HCFC-123 and HCFC-124 may be due to leakage during HFC-125 production"**
by Luke M. Western et al.
ACP manuscript egusphere-2025-3000

By:
*Kuijpers, Dr. Lambert J.M.*
Copernicus Office user ID 192458
(ORCID 0000-0002-0979-9694)

1.
The title "Increasing emissions of HCFC-123 and HCFC-124 may be due to leakage during HFC-125 production" is actually implying too much uncertainty, where it concerns the increase (amount of) emissions, and what the leakage from HFC-125 production could be. I would expect that certain readers would be of the opinion that the paper should not be written now, but once there is more certainty, at a later stage.

2.
One could also consider to change the title slightly, to "Impact of the leakage during HFC-125 production on the increase of HCFC-123 and HCFC-124 emissions". It does not mean that one has to give exact numbers for the leakage in Gg/yr (see the text of the article).

3.
The sections abstract, introduction, methods, results, discussion and conclusions have been very well written. For the methods and results sections, they consider all sources of emissions available, give background insights and the limits of how various issues should be considered.

There a few issues that I noted:
a. 58-62. "Both HCFC-123 and HCFC-124 are known to at least partly degrade in the atmosphere to trifluoroacetic acid (TFA) following their reaction with hydroxy".... Nice info, but **of less importance to the paper and its considerations**
b. 113. "NOAA's Global Monitoring Laboratory in Boulder, CO USA", **mention Colorado** as in line 122
c. 225-231. Prior knowledge of the production, $P_{i,t}$, is used to inform the estimation of the parameters of interest, utilising data reported to the United Nations Environment Programme. The production data reported to UNEP are the total production for all end-uses and, separately, as production for feedstock uses. To avoid double counting, we calculated the dispersive production by subtracting the feedstock production from the total production. **Could one please specify which data for which chemicals (feedstock) and for which years have been studied?**
d. I would be in favour of **phase-down and phase-out, with dashes**
e. 320. This is a very important sentence, and would deserve all info in all the sentences that follow (*Its emissions have not declined, and possibly increased, over 2018-2023, in line with an increase in HCFC-124 emissions over 2019-2023*). Lines 332-335 could be

elaborated further on what can actually be stated about the relationship between HCFC-123 (-124) and HFC-125 production.

f.  370 and further down. It starts "Using events where air conditioning units containing HFC-125 had leaked"….". The air conditioning application deserves more attention (starting with a better introduction, explaining various issues …). The largest use of HFC-125 in air conditioning is in the form of R-410A, a blend of HFC-125 and HFC-32 (roughly 50% each), and not HFC-125 as a pure HFC. Given the current attention on reducing the use of substances with high GWP, such as R-410A, and the possible replacement by pure HFC-32, a process that is already ongoing in many developed and some developing countries for many years, this may have much more impact on HFC-125 production than the reduction of production (and consumption) compliant with the Kigali schedule. This is also an aspect that needs mentioning in the conclusions.

g.  385-387. "Both HCFC-123 and HCFC-124 can break down in the atmosphere to form TFA, which can accumulate in aquatic bodies. The harmfulness of TFA is still uncertain. What is certain is that increasing emissions of HCFC-123 and HCFC-124 will lead to more TFA formation, and therefore accumulation in the environment". These sentences can be deleted here (in 385-387). They are not needed; the issue has been mentioned before.

h.  389. Given what has been mentioned in paragraph (f), this sentence "Yet production of HFC-125 is projected to increase in the coming years." needs more thorough analysis, together with references.

i.  395-406. The conclusions mention a number of issues correctly. The emissions of HCFC-123 are increasing, but it is difficult to say precisely what the source is. The emissions of HCFC-124 are increasing, but one cannot conclude from measurements where they are coming from. So, it leaves little room for hard conclusions, an issue that, together with the possibilities given in the title, may lead to experts giving as their opinion that this paper is, or will be, published at a too early stage.

j.  405: "An increase in monitoring stations around the world would help to better locate and understand the emission sources of HCFC-124". As a last conclusion, this is rather weak. The building of measurement stations will cost many years (investment possibilities), so good results will not be available until at some stage in the future, when many other atmospheric conditions may have changed. An IMPORTANT issue to mention here is how the trend of the future HFC-125 production will be, largely concerning future air conditioning trends, i.e., the production numbers for the market.

k.  In summary, the beginning and the end of the paper (title, intro, and conclusions) should be stronger. I.e., the way the title is formulated, and the main issues in the conclusions, specifics regarding the emissions of the two HCFCs, possible impacts on ozone and climate (warming), and future trends on HFC-125 (R-410A) production and consumption trends (maybe also its climate (warming) impact) compared to the two HCFCs the emissions of which are studied.

P.S. The draft needs a spellcheck, some commas can be added or deleted, some verbs should be conjugated in the singular, rather than the plural form, but that is a minor issue

One example (conclusions), where I have done a strikeout underline (conclusions)

Emissions of the hydrochlorofluorocarbons HCFC-123 and HCFC-124 have increased since 2019 despite a phasedown of their production and consumption for dispersive uses under the Montreal Protocol. Any increase in the production of HCFCs should only be for their use to produce other chemicals. Both HCFC-123 and HCFC-124 are intermediates in the production of the non-ozone-depleting greenhouse gas HFC-125, which has been used to  replace HCFCs for refrigeration and air conditioning and  fire suppression applications. Our analysis suggests that the increase in global emissions of HCFC-124 can be explained  by leakage during the production of HFC-125. We estimate this leakage rate as ~1.0% of HFC-125 production. Other sources of emissions cannot be ruled out. However, emissions of HCFC-123, another intermediate in HFC-125 production, also potentially increased with similar timing to HCFC-124. It is unclear from where the increase in emissions of HCFC-124 since 2019  is originating, as emissions in  Western Europe, the USA and East Asia have not increased since 2015, unlike the global emissions. Prior to 2019, emissions from  Western Europe, the USA and East Asia  were insufficient to explain the global changes over this period. An increase in monitoring stations around the world would help to better locate and understand the emission sources of HCFC-124

---

## Referee Comment (RC2)

Review of the paper "Increasing emissions of HCFC-123 and HCFC-124 may be due to leakage during HFC-125 production" by Western et. al.

This paper identifies sources of HCFC-123 and HCFC-124 emissions that point to their unexpected persistence in the atmosphere despite phase-out regulations under the Montreal Protocol. The work shows measurements of these species taken at several regional and global stations, which are combined with inverse modelling to estimate emission rates, especially from Asia, Europe, and the United States. The authors discuss that these emissions are by-products in the production process of HCFC-125, a non-ozone depleting substance, used in many applications instead of regulated compounds. Thus, the authors propose that current levels of HCFC-123 and HCFC-124 come from emissions generated during the production process of HCFC-125. In my view, this source identification is an important scientific contribution that fits within the scope of ACP and is a meaningful finding for future regulations of ODS. However, there are some points that require clarification prior to acceptance for publication:

Figure 1 (a, b) shows that the atmospheric abundance of HCFC-124 has always been greater in the northern hemisphere. At the same time, there are very few measurements in the southern hemisphere. Granted that most of the production is done in the northern hemisphere, it seems strange that this species never mixed completely in the atmosphere in 30 years. Could it be that a notorious lack of measurements in the southern hemisphere is introducing a bias in the calculation? Can authors offer some quantitative idea of how the global mean is affected by this aspect?

Authors conclude that HCFC-124 emissions have being increasing globally but the source region for excess emissions is not clear because emissions from Europe, US, and eastern Asia have not increased since 2015. However, looking at Figure 5, 6, and 7, this conclusion seems counterintuitive. Looking at the entire time of the trends presented in the figures, emissions have fallen consistently in Europe and the US, but they have increased in eastern Asia. Thus, it seems that one of the source regions of HCFC-124, precisely, is eastern Asia. Hence, there seems to be either an apparent contradiction or a tacit implication that the authors should clarify.

The previous point is also evident if the figures in the paper are seen as a whole. According to the authors, emissions of HCFC-124 from the US dropped to almost half the amounts in 2015 (0.5 to 0.26 Gg yr$^{-1}$). Emissions in Europe have dropped in line with phase out regulations. However, emissions of HCFC-124 from eastern China more than doubled in 15 years (0.2 to 0.45 Gg yr-1) and industrial regions are identified. It seems that the authors feel conservative at directly stating that eastern China is a source region. Could the authors comment on this?

It would be beneficial to the paper and the readership that authors include a summary of the HCFC-125, HCFC-124, HCFC-123 inventories or reported values from every region and contrast the inversion model results with the reported values.

The authors use inverse modeling to estimate emissions from source regions (Europe, US, and eastern Asia) for which they use a priori emission estimates. Did the authors consider using a priori values (even if arbitrary as done in this and other papers) to estimate/model potential emissions from other regions such as from Brazil in South America?

Looking at the entire trend in Figure 3a, it shows a global decrease of HCFC-124 from 1997 until 2019, when authors indicate that emissions begin to increase again. In Figure 3b, HCFC-123 emissions have continuously increased for the same time period, although with greater uncertainty. In the past 5 years or so, the speed of increase of HCFC-123 seems to have decreased. If both are byproducts in the production of HCFC-125, how is it that only HCFC-124 generally decreased while HCFC-123 generally increased? Although the authors discuss uncertainties in HCFC-123 and acknowledge the different trends, it is not entirely clear/convincing why both trends are so different. The authors should better clarify this portion of the paper.

Also, with respect to Figure 3a, since 2019 HCFC-124 emissions seem to go up again, which authors point to in the text. However, there are many bumps just like that (or bigger) in previous years in the entire trend and yet as a whole the trend is negative since about 99. Could it be that this "bump" is similar to the previous ones and the trend could still be negative overall?

The authors recognize that the global increase in HCFC-123 and 124 emissions does not currently represent a threat to the ozone layer recovery, but they present the important implication of whether other ODS or GHG could be emitted (or are being presently emitted) as intermediates from the production processes of fluorochemicals. This perspective is eye-opening. Could the authors offer some more discussion on potential substances or industrial processes that should be investigated more closely?

---

## Author Comment (AC1)

We thank the reviewer for their comments, which have improved the manuscript. Below we detail our responses to all comments in bold text.

Figure 1 (a, b) shows that the atmospheric abundance of HCFC-124 has always been greater in the northern hemisphere. At the same time, there are very few measurements in the southern hemisphere. Granted that most of the production is done in the northern hemisphere, it seems strange that this species never mixed completely in the atmosphere in 30 years. Could it be that a notorious lack of measurements in the southern hemisphere is introducing a bias in the calculation? Can authors offer some quantitative idea of how the global mean is affected by this aspect?

The continued interhemispheric difference is due to continued higher emissions in the Northern Hemisphere compared to the Southern Hemisphere. We have added the additional sentence at line 274 for clarity, "Persistent higher mole fractions in the Northern Hemisphere are indicative of continued higher emissions in the Northern Hemisphere compared to the Southern Hemisphere."

Authors conclude that HCFC-124 emissions have being increasing globally but the source region for excess emissions is not clear because emissions from Europe, US, and eastern Asia have not increased since 2015. However, looking at Figure 5, 6, and 7, this conclusion seems counterintuitive. Looking at the entire time of the trends presented in the figures, emissions have fallen consistently in Europe and the US, but they have increased in eastern Asia. Thus, it seems that one of the source regions of HCFC-124, precisely, is eastern Asia. Hence, there seems to be either an apparent contradiction or a tacit implication that the authors should clarify.

We have revised our language to make this clearer. In the discussion we now say "Therefore, based on the emissions estimates in this work, emissions from East Asia are insufficient to explain the entire global increase in emissions prior to 2015, and cannot explain the increase since 2015." We have changed the abstract to say "Emissions of HCFC-124 from western Europe, the USA and East Asia have either fallen or not increased since 2015 and together cannot explain the entire increase in the derived global emissions of HCFC-124."

The previous point is also evident if the figures in the paper are seen as a whole. According to the authors, emissions of HCFC-124 from the US dropped to almost half the amounts in 2015 (0.5 to 0.26 Gg yr-1). Emissions in Europe have dropped in line with phase out regulations. However, emissions of HCFC-124 from eastern China more than doubled in 15 years (0.2 to 0.45 Gg yr-1) and industrial regions are identified. It seems that the authors feel conservative at directly stating that eastern China is a source region. Could the authors comment on this?

We do not want to shy away from stating that China is a source region and have added the following at line 370 for clarity, "Emissions from eastern China are currently the largest of the regions studied and is therefore the largest known source region." However, as the remainder of the discussion in lines 380-388 express, our results show that emissions from eastern China are not solely responsible for the global

increase in emissions of HCFC-124 and cannot explain any of the global increase since 2016, given the uncertainties.

It would be beneficial to the paper and the readership that authors include a summary of the HCFC-125, HCFC-124, HCFC-123 inventories or reported values from every region and contrast the inversion model results with the reported values.

As we do not quantify regional emissions of HFC-125, we would not make this comparison in the text. We do not have independent bottom-up regional inventories of emissions of HCFC-123 and HCFC-124, with the exception of emissions of HCFC-124 from the US EPA. This is stated in the text already, "This estimate agrees with the bottom-up estimate by the US Environmental Protection Agency, who report that less than 500 tonnes of HCFC-124 (no exact value given) were emitted annually 2018-2022 (US EPA, 2024)". Given that no clear quantitative estimate is provided by the US EPA, further analyses are not possible.

The authors use inverse modeling to estimate emissions from source regions (Europe, US, and eastern Asia) for which they use a priori emission estimates. Did the authors consider using a priori values (even if arbitrary as done in this and other papers) to estimate/model potential emissions from other regions such as from Brazil in South America?

Given the lack of measurements in South America, we are unable to estimate emissions for this region.

Looking at the entire trend in Figure 3a, it shows a global decrease of HCFC-124 from 1997 until 2019, when authors indicate that emissions begin to increase again. In Figure 3b, HCFC-123 emissions have continuously increased for the same time period, although with greater uncertainty. In the past 5 years or so, the speed of increase of HCFC-123 seems to have decreased. If both are byproducts in the production of HCFC-125, how is it that only HCFC-124 generally decreased while HCFC-123 generally increased? Although the authors discuss uncertainties in HCFC-123 and acknowledge the different trends, it is not entirely clear/convincing why both trends are so different. The authors should better clarify this portion of the paper.

As you say, given the uncertainties in the derived emissions, it is very difficult to draw any quantitative conclusions about changes in emissions of HCFC-123 over time. We therefore choose not to discuss any apparent changes to the trend in the mean derived HCFC-123 emissions over time, beyond what is already included. We acknowledge the differences in the historical emissions of HCFC-123 and HCFC-123 at line 324, "The emissions trend of HCFC-123 since 1993 is uncertain and does not appear to follow that of HCFC-124, perhaps due to their different dispersive uses prior to their phase-down."

Also, with respect to Figure 3a, since 2019 HCFC-124 emissions seem to go up again, which

authors point to in the text. However, there are many bumps just like that (or bigger) in previous years in the entire trend and yet as a whole the trend is negative since about 99.

Could it be that this "bump" is similar to the previous ones and the trend could still be negative overall?

This is an important point that we had not discussed. We have added the following to the Discussion:

"The global mean emissions of both HCFC-123 and HCFC-124 have large interannual variability between some years. For HCFC-124, the increase after 2019 is of similar magnitude to the interannual variability in some years, particularly using the NOAA record. This may be due to random errors in the estimation procedure or changes in the atmospheric sink and other dynamic effects not considered in the global box model. For HCFC-124, we believe that the increase in emissions after 2019 is genuine, given that the increase in emissions is driven by a slowing in the decline of the background atmospheric abundances, which are measured directly and precisely, and the global mean is subject to less uncertainty than the derived global emissions. The result is also robust between the independent NOAA and AGAGE networks in both the abundance trends and emissions. If this impact was due to dynamical or loss related changes, this change would be apparent in other atmospheric compounds with loss process dominated by the hydroxyl radical."

The authors recognize that the global increase in HCFC-123 and 124 emissions does not currently represent a threat to the ozone layer recovery, but they present the important implication of whether other ODS or GHG could be emitted (or are being presently emitted) as intermediates from the production processes of fluorochemicals. This perspective is eye-opening. Could the authors offer some more discussion on potential substances or industrial processes that should be investigated more closely?

We do not have recommendations for potential substances or industrial processes that should be investigated more closely beyond those already discussed in the scientific literature, which we discuss in the introduction, "Various other CFCs and HCFCs have had persistent, or increasing, emissions (Adcock et al., 2018; Lickley et al., 2020; Vollmer et al., 2021, 2018; Western et al., 2023). The increase in the emissions of these CFCs and HCFCs has largely been attributed to their involvement in the production of other chemicals, most notably hydrofluorocarbons (HFCs) (Adcock et al., 2018; Kloss et al., 2014; Vollmer et al., 2015, 2018, 2021; Western et al., 2023), which is allowed under the Montreal Protocol."

---

## Author Comment (AC2)

We thank the reviewer for their comments, which have improved the manuscript. Below we detail our responses to all comments in bold text.

The title "Increasing emissions of HCFC-123 and HCFC-124 may be due to leakage during HFC-125 production" is actually implying too much uncertainty, where it concerns the increase (amount of) emissions, and what the leakage from HFC-125 production could be. I would expect that certain readers would be of the opinion that the paper should not be written now, but once there is more certainty, at a later stage. One could also consider to change the title slightly, to "Impact of the leakage during HFC-125 production on the increase of HCFC-123 and HCFC-124 emissions". It does not mean that one has to give exact numbers for the leakage in Gg/yr (see the text of the article).

Thank you for your suggestion. We have revised the title to "Impact of leakage during HFC-125 production on the increase in HCFC-123 and HCFC-124 emissions".

58-62. "Both HCFC-123 and HCFC-124 are known to at least partly degrade in the atmosphere to trifluoroacetic acid (TFA) following their reaction with hydroxy".... Nice info, but of less importance to the paper and its considerations

We understand that the conversion to TFA is not of immediate consequence to the ozone layer or climate but is nevertheless relevant for introducing the wider impacts of these HCFCs and to the Montreal Protocol. For example, Decision XXXV/3 at the Meeting of the Parties to the Montreal Protocol asked the Scientific Assessment Panel to lead research into "Early identification and quantification of any substances ... in particular those with high global warming potential, breakdown products of controlled substances and their alternatives that are very persistent, such as perfluoro- and polyfluoroalkyl substances, including trifluoroacetic acid."

113. "NOAA's Global Monitoring Laboratory in Boulder, CO USA", mention Colorado as in line 122

This sentence is now "The flasks are returned to NOAA's Global Monitoring Laboratory in Boulder, Colorado, USA for analysis on the Perseus GC-MS instrument."

225-231. Prior knowledge of the production, Pi,t, is used to inform the estimation of the parameters of interest, utilising data reported to the United Nations Environment Programme. The production data reported to UNEP are the total production for all enduses and, separately, as production for feedstock uses. To avoid double counting, we calculated the dispersive production by subtracting the feedstock production from the total production. Could one please specify which data for which chemicals (feedstock) and for which years have been studied?

The data that we were provided did not state which chemicals the feedstock were used. We have added/changed the following in the text (line 229), "The data do not

contain information on how the feedstock production was used. We assume that all reported HCFC-124 feedstock production was used in HFC-125 production and therefore is captured by HFC-125. Reported production in 2003 and 2004 was anomalously low (around two orders of magnitude smaller than in previous and following years) and thus we chose to begin our simulations in 2005, and production data are used from 2005-2023."

I would be in favour of phase-down and phase-out, with dashes

We use phase-down/phase-out as a noun and phase down/phase out a verb. We have ensured that this is consistent within the manuscript (there was one inconsistency).

320. This is a very important sentence, and would deserve all info in all the sentences that follow (Its emissions have not declined, and possibly increased, over 2018-2023, in line with an increase in HCFC-124 emissions over 2019-2023). Lines 332-335 could be elaborated further on what can actually be stated about the relationship between HCFC-123 (-124) and HFC-125 production.

We have expanded the discussion from what was previously line 332, which is now "Sources of emissions of HCFC-123 other than during HFC-125 production, including a bank of unknown size, will also contribute to the global emissions. A report projecting HCFC-123 bank emissions from 2002 onwards estimated that emissions from the bank could be between 1.6-5.6 Gg yr1 by the year 2015 (Ashford et al., 2004), compared to our estimate of 5.1  $\pm$  1.7 Gg yr1, noting that the date of the complete phase-out HCFCs under the Montreal Protocol was brought forward by 10 years in 2007. It is not known to what extent HCFC-123 is used to produce CFC-113a and other chemicals. We are unable to disentangle emissions of HCFC-123 from the bank and those from chemical production. As emissions of HCFC-123 in 2023 were larger than those of HCFC-124, and it is expected that less HCFC-123 intermediate will be emitted than HCFC-124 given its earlier fluorination step, it is likely that there is another substantial source of HCFC-123 in addition to HFC-125 production. Our analysis of HCFC-124 emissions using knowledge of production has allowed separation of emissions from the bank and from chemical-production. Given the short atmospheric record of regular measurements of HCFC-123, and the uncertainty in trends prior to this record, we are unable to draw any firm conclusions, simply observing that the timing of the increase in the global emissions of HCFC-123 and HCFC-124 is similar and that both are intermediates in HFC-125 production."

370 and further down. It starts "Using events where air conditioning units containing HFC-125 had leaked"....". The air conditioning application deserves more attention (starting with a better introduction, explaining various issues ...). The largest use of HFC-125 in air conditioning is in the form of R-410A, a blend of HFC-125 and HFC-32 (roughly 50% each), and not HFC-125 as a pure HFC. Given the current attention on

reducing the use of substances with high GWP, such as R-410A, and the possible replacement by pure HFC-32, a process that is already ongoing in many developed and some developing countries for many years, this may have much more impact on HFC-125 production than the reduction of production (and consumption) compliant with the Kigali schedule. This is also an aspect that needs mentioning in the conclusions.

The paragraph now begins, "Some HFC-125 installed in appliances may still be contaminated with intermediates or by-products that were not properly removed during the production process. Possible contamination should remain detectable if the composition of the installed refrigerant is measured. We do not have direct analyses of installed refrigerant composition; however, there are occasions in which the refrigerant in the air conditioning unit at measurement stations has leaked, and the composition is thus measurable. Using events where air conditioning units..."

We now add the additional information, now at line 421, "The production of HFC-125 is projected to increase in the coming years (Velders et al., 2022). However, a general transition to lower GWP refrigerants, such as HFC-32, may mean that the phase-down of HFC-125 is faster than scheduled under the Kigali Amendment."

385-387. "Both HCFC-123 and HCFC-124 can break down in the atmosphere to form TFA, which can accumulate in aquatic bodies. The harmfulness of TFA is still uncertain. What is certain is that increasing emissions of HCFC-123 and HCFC-124 will lead to more TFA formation, and therefore accumulation in the environment". These sentences can be deleted here (in 385-387). They are not needed; the issue has been mentioned before.

**This has been deleted.**

389. Given what has been mentioned in paragraph (f), this sentence "Yet production of HFC-125 is projected to increase in the coming years." needs more thorough analysis, together with references.

As mentioned in response to the earlier comment, we have changed this to "The production of HFC-125 is projected to increase in the coming years (Velders et al., 2022). However, a general transition to lower GWP refrigerants, such as HFC-32, may mean that the phase-down of HFC-125 is faster than scheduled under the Kigali Amendment."

395-406. The conclusions mention a number of issues correctly. The emissions of HCFC-123 are increasing, but it is difficult to say precisely what the source is. The emissions of HCFC-124 are increasing, but one cannot conclude from measurements where they are coming from. So, it leaves little room for hard conclusions, an issue that, together with the possibilities given in the title, may lead to experts giving as their opinion that this paper is, or will be, published at a too early stage.

As in our response to the earlier comment, we have now revised the title following your suggestion to "Impact of leakage during HFC-125 production on the increase in HCFC-123 and HCFC-124 emissions". Our conclusions state that "Our analysis suggests that the increase in global emissions of HCFC-124 can be explained from leakage during the production of HFC-125. We estimate this leakage rate as ~1.0% of HFC-125 production. Other sources of emissions cannot be ruled out." These conclusions remain valid.

405: "An increase in monitoring stations around the world would help to better locate and understand the emission sources of HCFC-124". As a last conclusion, this is rather weak. The building of measurement stations will cost many years (investment possibilities), so good results will not be available until at some stage in the future, when many other atmospheric conditions may have changed. An IMPORTANT issue to mention here is how the trend of the future HFC-125 production will be, largely concerning future air conditioning trends, i.e., the production numbers for the market.

**Please see the response to the following comment.**

k. In summary, the beginning and the end of the paper (title, intro, and conclusions) should be stronger. I.e., the way the title is formulated, and the main issues in the conclusions, specifics regarding the emissions of the two HCFCs, possible impacts on ozone and climate (warming), and future trends on HFC-125 (R-410A) production and consumption trends (maybe also its climate (warming) impact) compared to the two HCFCs the emissions of which are studied.

Thank you for this suggestion. We have reordered and adapted our discussion and conclusions, which should address the concerns raised here and previously. Our final paragraph is now, "The increase in emissions of HCFC-123 and HCFC-124 are likely to have a negligible impact on stratospheric ozone recovery at present, and little impact on the climate. The production of HFC-125 is projected to increase in the coming years (Velders et al., 2022). However, a general transition to lower GWP refrigerants, such as HFC-32, may mean that the phase-down of HFC-125 is faster than scheduled under the Kigali Amendment. There is no quantitively mandated degree to which production of substances controlled under the Montreal Protocol, or their emission, must be mitigated during the production of other chemicals (except for HFC-23). It may be that leakage rates of feedstocks, intermediates and by-products are higher than previously thought, suggesting that large quantities of ozone-depleting substances and potent greenhouse gases could be emitted from new fluorochemical production for many years to come."

P.S. The draft needs a spellcheck, some commas can be added or deleted, some verbs should be conjugated in the singular, rather than the plural form, but that is a minor issue

| We have tried our best to ensure that the spelling and verbiage is correct in the revised version. |
|----------------------------------------------------------------------------------------------------|
|                                                                                                    |
|                                                                                                    |
|                                                                                                    |
|                                                                                                    |
|                                                                                                    |
|                                                                                                    |
|                                                                                                    |
|                                                                                                    |
|                                                                                                    |
|                                                                                                    |